# Reviewing Competence in Practice: Reform of Continuing Professional Development for Irish Pharmacists

**DOI:** 10.3390/pharmacy7020072

**Published:** 2019-06-20

**Authors:** Mary-Claire Kennedy, Aisling Reast, Katherine Morrow, Frank Bourke, Claire Murphy, Richard Arnett, Catriona Bradley

**Affiliations:** 1School of Healthcare, University of Leeds, Leeds LS2 9JT, UK; m.c.kennedy@leeds.ac.uk; 2Irish Institute of Pharmacy, based at the Royal College of Surgeons in Ireland (RCSI), D02 FP84 Dublin, Ireland; aislingreast@appel.ie (A.R.); katherinemorrow@iiop.ie (K.M.); frankbourke@iiop.ie (F.B.); clairemmurphy@iiop.ie (C.M.); 3Royal College of Surgeons in Ireland (RCSI), D02 YN77 Dublin, Ireland; rarnett@rcsi.ie

**Keywords:** Continuing Professional Development, competence, competency, pharmacy, practice review, ePortfolio Review, Irish Institute of Pharmacy

## Abstract

There has been significant reform of the Continuing Professional Development (CPD) requirements for Irish pharmacists over the past five years. In 2015, a new system was established that includes quality assurance of practitioner engagement in CPD and quality assurance of practitioner competence. Pharmacists must now plan and document their learning activities in an electronic portfolio (ePortfolio) and they must participate in an ePortfolio Review process once every five-year period. A random sample is chosen each year to participate in a review of their practice for pharmacists in patient-facing roles. This paper provides an overview of the development and implementation of these quality assurance processes and it considers the outcomes that were observed in the first four years of implementation. By April 2019, almost 3000 pharmacists had participated in the ePortfolio Review process over the preceding three years, of which 96.2% demonstrated appropriate engagement in CPD. In the preceding two years, almost 200 pharmacists had participated in Practice Review, of which 97.5% have demonstrated the required level of competence across four competencies. All of the pharmacists who did not demonstrate the required level of competence in one or more competency area during Practice Review had previously demonstrated appropriate engagement in CPD through the ePortfolio Review process. This raises interesting questions regarding the use of engagement in continuing education (CE) or CPD as a surrogate measure for competence by professions.

## 1. Introduction

Pharmacy practice requires a combination of clinical, technical, and social skills to deliver safe, effective, and efficient pharmaceutical care. Developing and maintaining these skills, behaviors and knowledge is challenging due to the evolving nature of drug development, the ever-expanding evidence base, and caring for patients with often complex medical needs [1]. Pharmacists, like other healthcare professionals, must continuously strive to maintain and adapt their skills and knowledge to respond to these changes in their practice, thus necessitating a culture of lifelong learning and development [2].

Continuing Professional Development (CPD) is an ongoing process of learning that is reflective of the specific needs of the individual. Ongoing learning can be achieved through a variety of methods, including the completion of structured credit and non-credit bearing courses and unstructured ‘on the job’ learning [3]. It differs from continuing education (CE), which requires individuals to engage with a pre-defined educational programme that is not necessarily tailored to their practice or specific to their learning needs. When undertaking CE, an individual will not usually have an active role in deciding on the content and nature of the programme, meaning that it may have little application to their specific practice area [4]. The regulators of professions generally require evidence of participation in CE as a surrogate measure of CPD, often requiring evidence of hours of learning [5,6]. Within the pharmacy, some regulators have moved from such a CE model, instead requiring evidence of a more holistic approach to CPD. The self-deterministic nature of CPD is in-keeping with an andragogical model of learning, as described by Malcolm Knowles, in which motivated individuals identify and address their learning needs, which is the impetus for the desire to acquire this knowledge directly derived from their practice context [3,7].

An ongoing process of review of CPD can be helpful in determining an individual’s engagement with continued learning and the level of their professional competence. CPD review is also a mechanism for identifying those individuals that require further training or additional scrutiny. Asserting a regulatory requirement for professionals to undertake CPD and for an ongoing review of the level of this engagement, responds to the fact that a single determination of professional knowledge, skill, and behavior, such as that undertaken during undergraduate education or initial registration with the regulator, is insufficient for guaranteeing competence throughout a professional career [2]. The ultimate aim of such an exercise is to maintain and improve the safety and quality of care.

This paper provides an overview of the development and implementation of a new system of CPD recording and review for pharmacists in the Republic of Ireland, specifically describing the electronic portfolio (ePortfolio) system, review of ePortfolios, and practice review processes. Information regarding the development and implementation of the accreditation processes (assurance of Provision) is not reported here. 

## 2. Institutional Leadership and Policy Reform

Regulatory bodies have a duty to uphold professional standards to ensure that the care that is delivered to the public is safe and effective [2]. In recent years, pharmacy regulators in many jurisdictions have assumed greater oversight of CPD that is undertaken by registrants. In many cases, the regulators have redefined the way in which CPD is conducted and recorded, as well as formalizing and overseeing a process of review of CPD as a means of quality assuring registrants [2,8]. The Pharmaceutical Society of Ireland (PSI) is the regulator for all pharmacists (over 6000) in the Republic of Ireland and it is responsible for the registration of pharmacists, pharmaceutical assistants, and retail pharmacy businesses [9]. The PSI is also responsible for establishing and reviewing the standards for pharmacy education, including the accreditation of education providers [8]. The PSI commissioned a review of CPD models in 2010 with the aim of recommending a revised approach to CPD recording and review for pharmacists as part of their role in relation to maintaining education and professional standards [8]. The CPD report (2010) recommended that the model of Practice Review that was developed by the Ontario College of Pharmacists be used, being both feasible to implement in Ireland and sufficiently rigorous [8]. This model was adapted by the Irish Institute of Pharmacy (IIOP) for Irish requirements. Prior to this, each pharmacist retained complete autonomy in managing their ongoing learning and self-declared engagement with CPD at re-registration annually. There was no external review of CPD. 

One recommendation from the PSI CPD report (2010) was the establishment of the IIOP, an independent, to oversee the implementation of this new approach to recording and reviewing CPD [10]. The centralized organization of CPD review, coordinated and executed by the IIOP, helps to ensure that the assessment processes are tightly controlled and homogenous, which is a challenge for larger jurisdictions where responsibility is commonly devolved to regional centers [2]. The IIOP operates at “arm’s length” from the PSI whilst funded by, and reporting to, the PSI, meaning that data that is held by the IIOP, such as learning records, is not shared with the PSI. The IIOP have actively marketed their role to the wider pharmacy profession and they have an unambiguous leadership function in relation to CPD. This enables pharmacists to feel comfortable in identifying, recording, and addressing learning needs without the fear of disciplinary action. In fact, the statutory instrument states that the contents of a pharmacist’s ePortfolio are under the “*absolute control”* of the pharmacist [11]. 

Another notable recommendation arising from the CPD report (2010) was that a new system be established, which would verify the standards for CPD across the areas of provision (quality and relevance of the CPD activities being delivered by training providers), engagement (engagement of individual professionals in CPD), and overall competency of professionals (the competencies maintained or developed as a result of engaging in CPD activity). This required the IIOP to develop, establish, and maintain the following systems:-Accreditation systems for training programmes to provide quality assurance of provision-An electronic portfolio (ePortfolio) system, available to all pharmacists for managing and recording of their CPD activities. This was to include a tool for self-assessment of learning needs-An ePortfolio Review system, to provide quality assurance of engagement in CPD-A Practice Review system, to provide quality assurance of overall competency

The recommendations of the CPD report relating to the IIOP and each aspect of the CPD system (accreditation, ePortfolio, ePortfolio Review, and Practice Review) were subsequently formalized in a statutory instrument, which came into effect in 2015 [11]. In that instrument, it is specified that CPD must be *“systematic, self-directed, needs based and outcomes-focussed, based on a process of continual learning and development with application in… professional practice as a pharmacist*”. This represents a holistic interpretation of CPD and considerably varies from the more usual inputs-based requirements, where the accumulation of hours or points of approved learning are generally accepted as evidence of engagement in CPD.

Pharmacists are required to use an *“electronic CPD portfolio, established and maintained by the Institute for use by pharmacists”* [11]. They must provide evidence that they are using the ePortfolio and that they are satisfying the stated requirements for CPD, as defined above, as part of an ePortfolio Review process. Therefore, the ePortfolio tool and the ePortfolio Review process that were developed by the IIOP needed to enable pharmacists to meet these requirements.

Practice Review is defined in the statutory instrument as a “*direct evaluation, conducted by the Institute* [the IIOP]*, of the knowledge, skills and judgment of the pharmacist, against a standard established in consultation with peer pharmacists practising in patient-facing roles, having regard to the Core Competency Framework for Pharmacists, with particular reference to those competencies dealing with patient care, including clinical knowledge, the ability to gather and interpret appropriately information from and about patients, patient management and education and communication (including counselling) skills*” [11]. The Practice Review process that was developed by the IIOP needed to enable pharmacists to meet these requirements. 

Each of these systems needed to align with the PSI’s Core Competency Framework (CCF) (2013). The CCF details the skills, knowledge, and behaviors that should be demonstrated by all registered pharmacists, regardless of their area of practice [12]. The CCF is structured around six core domains: professional practice, personal skills, supply of medicines, safe and rational use of medicines, public health and organization, and management skills. A series of competencies are mapped to each of these six domains and a number of behavioral statements are attributed to each competency (Figure 1).

## 3. Development and Implementation of the ePortfolio

The IIOP was formally established in August 2013 and it became operational in March 2014 once appropriate staffing, accommodation, governance, and operational systems and processes had been established. One of the early priorities was to engage the pharmacy profession in the development of the new system. The IIOP recruited over 35 pharmacists through an Expression of Interest process to form a Peer Support Network. These pharmacists formed a valuable conduit between the profession and the IIOP. Training was provided to members of the network, who subsequently facilitated 60 meetings around the country during 2014. During these meetings, general information was presented on CPD, and an explanation of the new CPD system was provided before seeking attendees’ insights on the development of the new systems. Over 1800 pharmacists attended these meetings and provided rich insight into what they required from an ePortfolio system in terms of appearance, usability, and functionality. The examples included the demand for mobile responsiveness, easy navigation, use of self-explanatory icons, use of white space, and the ability to attach supplementary files to cycles (such as photos, videos, and documents). This, combined with the regulatory requirements, enabled the IIOP to develop a system specification and to engage an external company to develop a system. 

The IIOP ePortfolio was made available to pharmacists through the IIOP website in April 2015, following multiple rounds of user acceptance testing and refinement by the Peer Support Network. The ePortfolio tool guides pharmacists through the process of competence improvement by enabling them to plan, document, and reflect on their learning in the form of CPD cycles. CPD cycles are structured around a five-step model, with reflection being essential to actively rationalize the significance of each of the stages (Figure 2). The dynamism and self-directivity that are assumed to be core values when planning, undertaking, and recording a learning activity embody the principles of andragogical learning and are born out in the five-step cycle [13].

The ePortfolio also incorporates a Core Competency Self-Assessment Tool (CCSAT) that enables pharmacists to self-assess against the PSI’s Core Competency Framework, which satisfies the regulatory requirement for regular self-assessment. The CCSAT helps pharmacists to identify learning needs that may not have been previously apparent to that individual and that may help to orient the pharmacist’s ongoing learning.

## 4. Development and Implementation of the ePortfolio Review Processes

Irish pharmacists are required to engage in ePortfolio Review once in every five- year period. A cohort of pharmacists are selected at random from the register by the PSI each year and this list is then issued to the IIOP, who co-ordinate ePortfolio Review. Selected pharmacists are invited to submit a selection of cycles from their ePortfolio, which are referred to as an extract, each year. There is one initial postal communication, after which all communication is electronically conducted. One further postal communication is issued in cases where a pharmacist does not submit an extract within the initial submission period. 

The IIOP developed and established an ePortfolio Review process, which enables pharmacists to submit extracts through the online portal of the IIOP website when requested to do so. Extracts are subject to review through two processes; review against System Based Standards and Peer Review (Figure 3). The standards that underpin both of these processes are defined by a group of pharmacists, from a range of practice backgrounds, who form a standard-setting group. The standards are agreed by the standard-setting group for each year of ePortfolio Review. In total, there are eight System Based Standards, the criteria of which are designed to check that pharmacists have identified their learning needs and have addressed these by engaging in a range of learning activities (which should include a mix of formal and non-formal learning approaches), with due consideration to the impact of this learning on their professional practice. For the 2018/19 ePortfolio Review, it was possible to meet the System Based Standards by submitting an extract that is composed of six cycles, including four cycles created in 2018 (one of which must be created following self-reflection against the Core Competency Framework), two cycles that were created in the previous four years (2014–2017); at least two cycles were required to start at the self-appraisal stage. 

The first stage of ePortfolio Review, the review against the System Based Standards, is an automated process that is executed through the ePortfolio portal on the IIOP website. Automation of this process standardizes and enhances the reliability of this stage of the review, and is possible as it involves the application of several defined rules to an extract of cycles that was submitted by a pharmacist.

Peer Review is the second phase of the ePortfolio review process. This stage of the review primarily reviews the nature and relevance of the information that was provided within the various stages of the five-step cycle. This phase of review is carried out by Peer Reviewers, who electronically access their allocated cases. Peer Reviewers are pharmacists who are recruited through an Expression of Interest process to the pharmacy profession and who have undergone specific training in conducting the review and providing appropriate feedback. 

All of the pharmacists who fail to meet the System Based Standards are automatically referred for Peer Review. As a quality assurance mechanism, a proportion of extracts that have satisfied the System Based Standards are also referred for Peer Review. A minimum of 20% of submissions undergo Peer Review each year. Pharmacists whose extracts have satisfied the System Based Standards and are not selected for Peer Review undergo no further scrutiny. There are two review periods. Following the first review period, pharmacists who have not satisfied the standards (System Based, Peer Review, or both) receive personalized feedback on their submission and they are afforded a further opportunity to re-submit additional cycles. Following the second review period, pharmacists receive one of four outcomes: “Standard Met”, “Standard Not Met (Year 1)”, “Standard Not Met (Year 2)”, or “Non-Engagement”. Pharmacists with an outcome of “Standard Met” are provided with certification of having met the required standards. Pharmacists with an outcome of “Standard Not Met (Year 1)” are entered into the next year’s review, thus affording them with another opportunity to demonstrate their engagement with CPD. Pharmacists who do not satisfy the standards for the second year receive an outcome of “Standard Not Met (Year 2)”, and these pharmacists are referred to the PSI. Pharmacists are deemed to be “Non-Engagers” if they have not engaged in the ePortfolio Review process, and these too are referred to the PSI. 

## 5. Practice Review: Competence Review of Patient-Facing Pharmacists

All of the registered pharmacists with patient-facing roles, such as those working in community or hospital sectors, are eligible to be randomly selected from the professional register by the PSI to participate in Practice Review. Part 5, Rule 12 (7) of the PSI (Continuing Professional Development) Rules 2015 states that

*“….practising in a “patient-facing role” means carrying out the role of a pharmacist in the delivery, or the oversight of the delivery, of care and services to members of the public, including patients, whether in a retail pharmacy business or in a pharmacy department of a hospital, or any other relevant location including on a casual or occasional basis, and includes the role carried out by a superintendent pharmacist, a supervising pharmacist and any other registered pharmacist engaged or employed in a retail pharmacy business or in the pharmacy department of a hospital”.* [11]

The statutory instrument states that Practice Review must determine the professional competence of the pharmacist in the following areas: clinical knowledge, the ability to gather and interpret appropriately information from and about patients, patient management, and education and communication (including counselling) skills [11]. The Review is composed of two elements: Standardized Pharmacy Interactions (SPIs) and a Clinical Knowledge Review (CKR). A Practice Review is conducted over four days each year, on a Saturday and Sunday, in April and October. Thirty-six pharmacists are called by the Regulator (PSI) on each of the days, 144 pharmacists each year. This represents approximately 4% of pharmacists on the register who have self-declared, at the point of annual re-registration with the PSI, where they work in a patient-facing role [14]. 

SPIs involve interactions with simulated patients. The scenarios, of which there are seven in total, do not require the pharmacist to dispense any medication, rather to respond to a medication-related issue or query presented by a ‘patient’ under their care. Pharmacists are allocated a consultation area, where they remain for the duration of the SPIs, and to whom each simulated patient presents, in turn. Each simulated patient has been trained in a specific case, and a Practice Reviewer accompanies it to each station. The Practice Reviewers have been trained in the use of checklists to objectively record the pharmacists’ performance in the following areas: gathering and interpreting information from and about patients, patient management, and education and communication (including counselling) skills [11]. 

The CKR, completed on the same day as the SPIs, is a computer-based assessment, whereby the pharmacist is presented with 18 clinical scenarios, on which a series of Single Best Answer (SBA) Multiple Choice Questions (MCQs) are posed. Pharmacists have access to any online resource that they wish, excluding social media fora, as well as paper-based resources. The CKR is designed to provide insight into the ability of pharmacists to answer questions on clinical scenarios, in an open book assessment.

While, on the surface, it may appear that SPIs are Objective Structured Clinical Examinations (OSCEs) by another name, this is not entirely accurate. An important step in the development of an OSCE is the mapping of each of the stations in the assessment to the intended learning outcomes of the course. However, in the case of SPIs, a formal taught course has not been delivered to pharmacists prior to participating in Practice Review. For this reason, scenarios have been developed by a panel of peers that are reflective of practice, and based on the competencies within the CCF dealing with patient care, including clinical knowledge, the ability to gather and appropriately interpret information from and about patients, patient management, and education and communication (including counselling) skills [11]. The clinical areas to be included in the Practice Review are identified by involving over 130 pharmacists, working in patient-facing roles in both hospital and community, in the development of a blueprint of the most commonly occurring clinical topics in practice. Every part of the process is peer-led, including blueprint development, question and case writing, question and case review, quality review, Angoff standard-setting group, practice reviewers, and practice review board.

## 6. Overview of Review Participation and Outcomes

Nearly 3000 pharmacists had been selected to participate in ePortfolio Review since 2016. Below, Table 1 provides a summary of outcomes for each review period.

Two-hundred pharmacists have attended for Practice Review. Table 2 below provides a summary of outcomes for each Practice Review event.

All of those who with an outcome of “Further Review Required” in Practice Review have received an outcome of “Standard Met” for ePortfolio Review.

## 7. Competence Development: Linking CPD to Professional Practice

Enhancing professional competence is dependent on the development of habitual, self-aware, and conscious practices [15]; and, the Irish system that has been developed would appear to achieve this in a number of ways. 

The ePortfolio tool supports pharmacists in self-assessment, planning, recording, and reflecting on their learning activities. ePortfolio systems have several advantages over traditional paper-based approaches to logging learning activities. Pharmacists can begin to populate a learning cycle within their ePortfolio as the learning is being undertaken and edited or enhanced the information in the cycle at a later time. Links and documentary evidence can be added to the electronic record, which might provide further evidence of the learning or activity that has been completed by the pharmacist. The domains and competencies from the CCF are listed alongside the five-step cycle, so that pharmacists can map the relevant competencies to the cycle. This aspect of the ePortfolio system is beneficial, as it ensures that the pharmacist is required to proactively consider their learning in the context of competence development, providing further meaning and context for their professional practice [16].

The ePortfolio Review provides quality assurance of the levels of engagement in CPD across the profession. The review of paper and electronic portfolios are common features of CPD assessment within various professions internationally and they are an efficient, cost-effective method of auditing registrants’ engagement with CPD [17]. The requirements for content and structure of portfolios of course vary, some regulators require mapping of evidence to a defined competency framework, while others require attendance and the logging of learning at a minimum number of structured learning events [17]. Similarly, there are differences in the frequency and extent of review of portfolios during the audit process. Such reviews tend to focus on engagement with learning activities (CE or CPD) and provide no assurance of competence, irrespective of the approach. 

The process of Practice Review offers an advantage over ePortfolio Review, in that it considers the professional competence of the pharmacist from an objective viewpoint. However, this does not necessarily reflect professional performance, as it does not capture the day-to-day practice behaviors of the pharmacist [18]. Competence and performance are not interchangeable concepts. Competence relates to the knowledge, skills, and behaviors that an individual should possess to capably execute a specific task in theory [19]. Competence also transcends these practical considerations that incorporate affective attributes, such as decision-making and beliefs, which are integral to the completion of professional tasks [20]. Performance relates to an individual’s execution of a task in the environment, in which that activity usually takes place. It recognizes the situated variables that affect how an individual may undertake a task, such as interaction with other staff members, the morale and culture of the healthcare institution in which they are employed, and the physical work setting [21]. 

The hierarchical distinction between competence and performance is effectively modelled within Miller’s Pyramid, where these concepts are labelled as ‘know how’ and ‘shows how’, respectively (Figure 4) [22]. 

The Dreyfus and Dreyfus model of skills acquisition (1986) also conceptualizes professional skills development along a continuum; commencing at the novice stage, the individual progresses to the level of expert over time as their clinical skills and knowledge strengthen; competence is the midway point of skills acquisition according to this model (Figure 5). The Dreyfus model argues that performance is implicit at each stage of skills development, unlike Miller’s Pyramid, in which performance is a discrete stage. Performance strengthens over time, with the quality of performance only determined when the execution of the skill is observed, this is not necessarily related with the career stage of the individual. Training and education facilitates the attainment of competence, but advancement beyond this stage to proficiency and the level of expert is only achieved by immersion in practice, during which time the individual consciously exercises their skill or knowledge appropriate to that context [24].

Regardless of the conceptual framework that is used to describe the arc of competence development, it is commonly accepted that good performance cannot occur in the absence of competence, they are not mutually exclusive concepts; that is, the individual must first be competent in order to perform a task in practice [22,23,25]. Traditional performance assessments require the observation of the pharmacist in their practice setting, as they engage with patients and colleagues [18,26]. Trained independent reviewers are required to conduct such an assessment in order to ensure fairness and transparency. However, ensuring the reliability and validity of a performance assessment is particularly challenging due to the unpredictable and varied nature of the work that is undertaken by pharmacists in practice. For example, one day in practice might be more challenging than another due to a high volume of prescriptions or complex patient or prescriber queries. Furthermore, the nature of pharmaceutical care might differ from one pharmacy to another. Therefore, competency-based assessments, which were conducted at intervals in controlled environments, such as Practice Review, are of use in predicting the performance of the individual in practice. 

Considering competence as a proxy measure of performance attracts many criticisms in the literature, reflecting the theoretical gap between the two concepts; as inference must be derived from the outcomes of a competence assessment regarding the individuals’ likely performance in practice [27]. However, it has been noted that, typically, all assessments involve a degree of inference; furthermore, the practicalities of a sufficiently robust assessment of performance means that competence review is more accessible [28]. The simulated nature of Practice Review affords certain advantages, the distractions of a busy workplace are removed and ensures that candidates are not disadvantaged by differences in the environment. Simulated patients, who consistently and concisely portray the case for each pharmacist, further enhance the reliability and validity of the assessment [29,30]. Critics of assessment in simulated environments have suggested that, given the artificiality of the setting, it is not possible to determine the true performance capabilities of an individual, again highlighting the gap between the concepts of competence and performance [31]. However, it can be argued that having complete control of the environment is preferable to providing a homogenous experience for all pharmacists.

MCQs are a widely used format for written assessments, being popular due to their reliability, objectivity, and efficiency [32]. The pharmacist is not required to engage with a taught syllabus prior to the assessment, as there is no curriculum underpinning Practice Review. Practicing pharmacists, to resemble clinical scenarios that are commonplace in practice, develop the questions within the CKRs, and the pharmacists have access to the same information resources that they would have in their workplace. Therefore, pharmacists are addressing clinical issues much as they would in practice. It can also be argued that resolving clinical issues is a process that is less influenced by the variability of the practice setting. Consequently, there is perhaps little difference between clinical competence demonstrated by the pharmacist in completing CKRs and the actual performance of the pharmacist in practice. 

It is necessary to acknowledge the potentially negative impact of test anxiety on the performance of pharmacists during Practice Review, despite the merits of both the SKIs and CKRs [33]. The heightened emotional and cognitive burden associated with assessment might mean that a pharmacist responds to a clinical scenario in a way that they ordinarily would not in their own practice setting. Unfortunately, anxiety will be a feature of any assessment model that is used to determine the knowledge or behaviors of an individual.

## 8. Instituting Changes in Recording and Reviewing CPD and Professional Competence: Challenges and Facilitators

Transition to an entirely new model of CPD and associated review necessitates a cultural shift and engagement from the professionals that are involved. The literature indicates that healthcare professionals concede that review processes prompt them to reflect on their professional conduct, although they are uncertain of the impact of this on their performance in everyday clinical practice [34,35]. There is also agreement that an appropriately structured assessment does succeed in identifying registrants that are unfit to remain in practice, thereby maintaining and improving the quality and safety of care that are provided by that professional group [36]. The apprehension of the profession in the face of such a significant change demands to be recognized and addressed, despite the acknowledgement that review of ongoing learning has positive elements. Effective institutional leadership, which motivates a cultural and attitudinal shift among key stakeholders, combined with coherent evidence-based policy reforms, are essential for successfully embedding changes to CPD requirements for a professional group [37]. 

The central role of registered pharmacists from a range of practice backgrounds in many aspects of the ePortfolio and Practice Review processes, including the development of standards against which ePortfolio cycles are considered, acting as Peer Reviewers, developing SPIs and CKRs, as well as acting as reviewers for these assessments, is central to the acceptability and validity of this new approach to CPD. Pharmacists who are working in practice have a pragmatic and realistic understanding of the necessary skills, knowledge, and behaviors that characterize a competent practitioner, and are therefore best-placed to define standards and review their peers.

The IIOP are visible and accessible to the profession providing face-to-face and online information giving sessions on a variety of topics, including the background to CPD for pharmacists for those who are newly registered with the PSI, engaging with ePortfolio Review, and undertaking Practice Review. The Peer Support Pharmacist network, which has increased in size from 35 to 70 since its inception, not only facilitates the dissemination of information about CPD requirements to the wider profession, but also ensures that individuals who are also engaging with these CPD reforms provide this information. The wider profession may be more receptive to such horizontal information sharing, between peers and colleagues, than vertical information sharing about CPD requirements directly emanating from the PSI or a direct employee of the IIOP.

The system is still in its infancy, with only three years of ePortfolio Review and six Practice Reviews events completed at the time of writing. As the system matures, interesting trends and insights are likely to emerge, and remediation processes will need to be further developed. However, in Ireland, we say “tús maith, leath na hoibre”, which means that a good start is half the work, and it would seem that the new CPD system has had a good start. 

## 9. Conclusions

Education and training reforms, such as changes to CPD requirements for healthcare professionals, require careful planning and execution to have any prospect to being of benefit to patients and the professional group involved. Being considered as a whole, a multi-method approach to a review process facilitates the consideration of competence through various lenses and mitigates against the deficiencies of each individual method. The IIOP have successfully engaged with pharmacists, enabling them to meet the CPD requirements that are defined in legislation. Successfully implementing sustainable CPD reforms is certainly the first step towards enhancing the quality of care, although it is a pharmacist’s own professional responsibility to translate competence into a meaningful impact on performance in practice. 

## Figures and Tables

**Figure 1 pharmacy-07-00072-f001:**
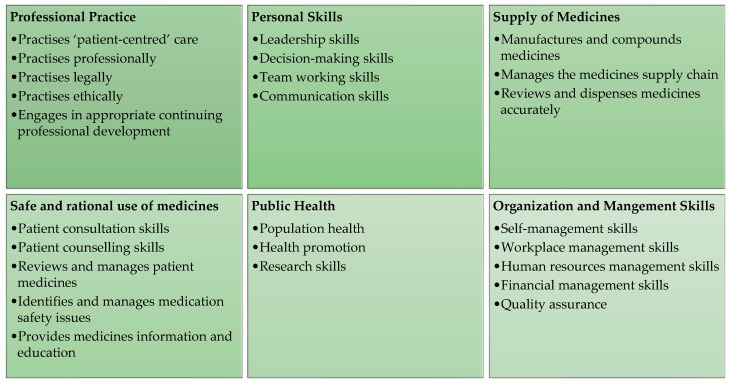
The Core Competency Framework for Pharmacists: Domains and Competencies.

**Figure 2 pharmacy-07-00072-f002:**
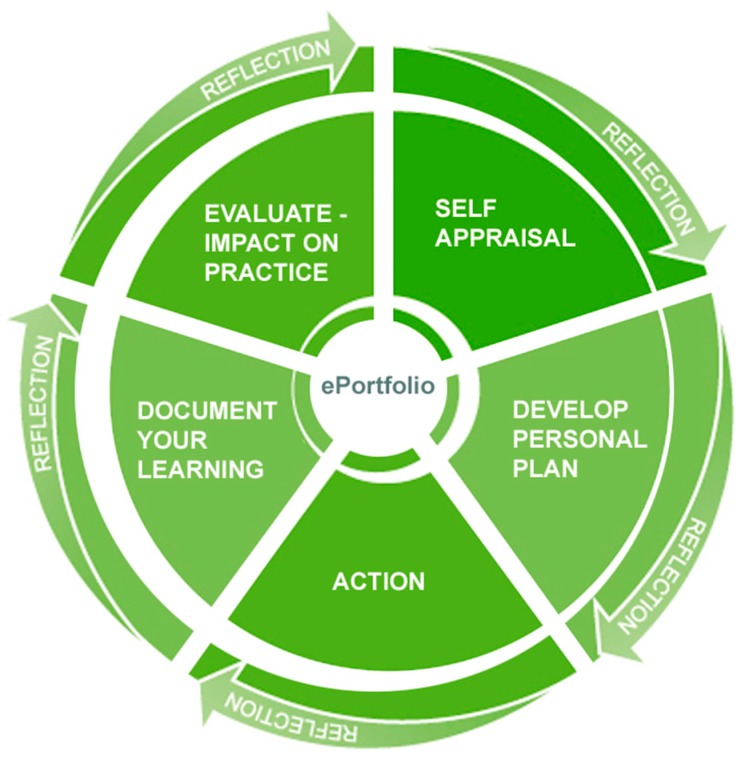
Irish Institute of Pharmacy (IIOP) Five-stage Continuing Professional Development (CPD) cycle.

**Figure 3 pharmacy-07-00072-f003:**
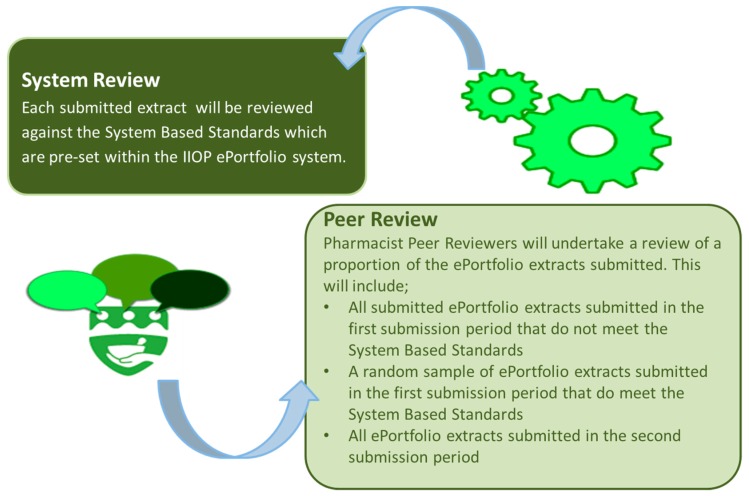
Overview of ePortfolio Review.

**Figure 4 pharmacy-07-00072-f004:**
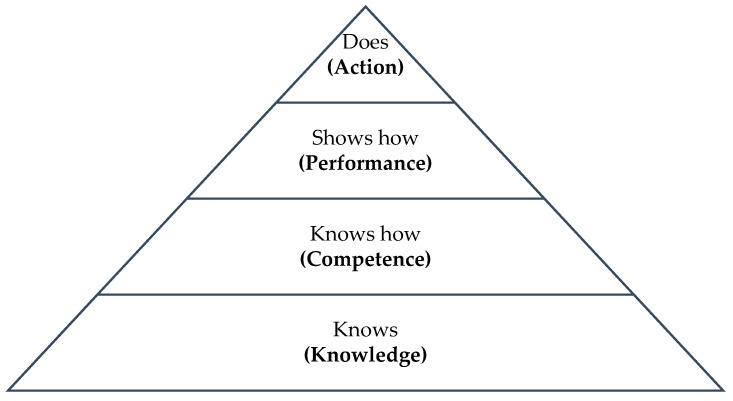
Millers Pyramid: Framework for Clinical Assessment [23].

**Figure 5 pharmacy-07-00072-f005:**
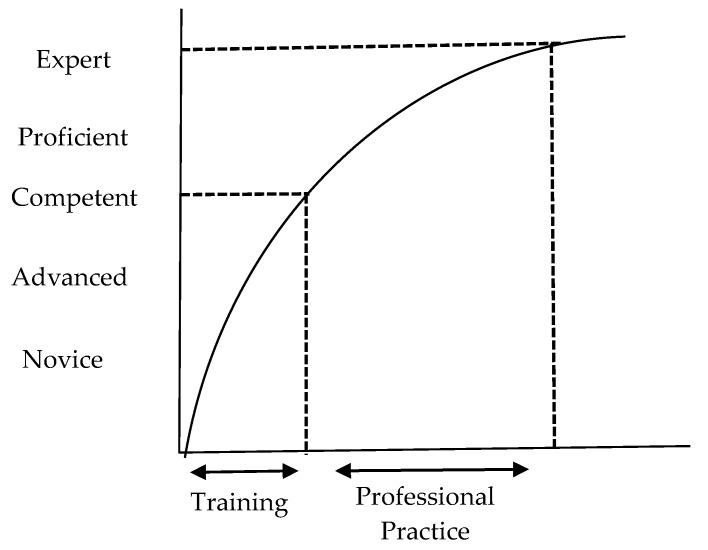
Skills acquisition curve adapted from ten Cate et al. [24].

**Table 1 pharmacy-07-00072-t001:** ePortfolio Review outcomes.

Review Year	Pharmacists Selected n	Pharmacists Participating n (%)	Outcome Standard Met n (%)	Outcome Standard Not Met (Year 1) n (%)	Outcome Standard Not Met (Year 2) n (%)	Outcome Non-Engagement n (%)
2016/17	258	243 (94.2%)	233 (90.3%)	10 (3.9%)	N/A	15 (5.8%)
2017/18	1246	1217 (97.7%)	1202 (96.5%)	15 (1.2%)	0	29 (2.3%)
2018/19	1338	1311 (98.0%)	1300 (97.2%)	10 (0.7%)	1 (0.1%)	27 (2%)

**Table 2 pharmacy-07-00072-t002:** Practice Review outcomes.

	April 2018	October 2018	April 2019
Pharmacists selected (n)	65	68	67
Pharmacists participating (n (%))	64 (98.4%)	67 (98.5%)	66 (98.5%)
Demonstrated competence (n (%))	63 (96.9%)	64 (94.1%)	64 (95.5%)
Further review required (n (%))	1 (1.5%)	3 (4.4%)	2 (3.0%)

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
