# Peer review of "Reviewing Competence in Practice: Reform of Continuing Professional Development for Irish Pharmacists"

_pharmacy, 2019, doi:10.3390/pharmacy7020072_

Round 1
Reviewer 1 Report
Paper Overview
The paper discusses the need for a change in the approach to CPD for pharmacists in Northern Ireland, the development of the infrastructure to enable a revalidation process to be implemented and the revalidation process to date. This includes the development of a portfolio to enable pharmacists to record their CPD and an assessment process for the portfolio records that utilises both a system marking and pharmacists review process. In addition, for those in patient facing roles, there is also a requirement for clinical skills to be assessed ensure that the pharmacists are both competent and able to demonstrate performance in practice situations. The paper provides a further contribution to the ongoing debate relating to ensuring the ongoing competence of pharmacists and the quality of their professional practice.
Strengths
The paper is clear about the process of the NI review process and how this is undertaken.
It provides a useful and informed discussion of the measurement or assessment of competence and performance.
The model suggested is of interest to all those who are engaged in trying to ensure that pharmacists in their country undertake adequate ongoing development to ensure the quality of the pharmacy professional workforce.
Weaknesses
The authors refer to andragogy in a couple of places in the paper without referencing the source of this theory of learning.
It could be made clear what the outcome is for those pharmacists who do pass the System Based Standards Review, and are not selected for the Peer Review.
The paper states the pharmacists self-identify if they are in a patient facing role but no details are given of how this is done and how often they are asked to update the process.
The tables in the paper are highly populated with data and represent a number of stages of review, and take some time to understand and follow.
The authors briefly discuss the possible draw backs of simulated environments but then state that compete control over the environment in this case is preferable, but the reason for this is not clearly elucidated.
Similarly, they comment that the MCQs will result in “little difference between the clinical competence of pharmacists in completing CKRs, and the actual performance of the pharmacist in practice” which seems to ignore issues relating to exam anxiety and the pressure that is experienced by some individuals in such a situation.
Specific comments
Line 25 – unclear what process if being referred to with regards to engagement in CPD
Line 47-48 it would be appropriate to include references to some examples from other regulators
Line 80 – uses ILLOP abbreviation before the ILLOP is introduced in line 85
Author Response
Thank you for your comments. We have taken them into account and edited the manuscript in accordance with your suggestions.
The authors refer to andragogy in a couple of places in the paper without referencing the source of this theory of learning.
Reference has now been made to Malcolm Knowles (Page 2 Line 53) who initially developed the theory around andragogy and a reference added to support this point.
It could be made clear what the outcome is for those pharmacists who do pass the System Based Standards Review, and are not selected for the Peer Review.
A sentence has been added (Lines 240-241), to clarify this point
The paper states the pharmacists self-identify if they are in a patient facing role but no details are given of how this is done and how often they are asked to update the process.
Lines 298-304 have been added to clarify this point
The tables in the paper are highly populated with data and represent a number of stages of review, and take some time to understand and follow.
We have revised the tables to enhance presentation and simply interpretation for readers
The authors briefly discuss the possible draw backs of simulated environments but then state that compete control over the environment in this case is preferable, but the reason for this is not clearly elucidated.
Line 419 have been revised to further explain the advantages of a simulated environment in this case.
Similarly, they comment that the MCQs will result in “little difference between the clinical competence of pharmacists in completing CKRs, and the actual performance of the pharmacist in practice” which seems to ignore issues relating to exam anxiety and the pressure that is experienced by some individuals in such a situation.
Thank you for highlighting this point which had not been discussed in the paper. Lines 431-437 have been added to acknowledge the issue around exam anxiety which might affect the performance of pharmacists during Practice Review.
Specific comments
Line 25 – unclear what process if being referred to with regards to engagement in CPD
This sentence has been altered to provide clarity
Line 47-48 it would be appropriate to include references to some examples from other regulators
References have been included to support this statement (References 5 & 6)
Line 80 – uses ILLOP abbreviation before the ILLOP is introduced in line 85
This has been revised
Reviewer 2 Report
An interesting paper outlining the introduction of a new system to assure practitioner engagement with CPD and practitioner competence in Ireland. Whilst this is not a new concept on a global scale, this does represent significant development to the competence assessment of pharmacy practitioners within Ireland and therefore is of general interest.
In general, the paper was well presented and flowed logically with accurate description of the developments. As a reader, I did struggle to understand the process of ePortfolio review against the system based standards and would suggest that this needs described in further detail to aid understanding (194-197).
I feel that the paper would also benefit from a diagrammatic overview of the process, such as a flow chart. There is a lot of text used to describe the process and a diagram would help to summarise the processes and to also aid the understanding of those who prefer to learn visually.
The paper does require review to ensure that it conforms with stylistic protocol. Examples that I have noted include:
(i) the abbreviation IIOP is used in line 80 without prior reference to the full name of the Centre.
(ii) in line 134, organization is spelt with a z whilst an s is used within Figure 1
(iii) within the references, the use of capital letters is sometimes missing eg. Reference 4, 15. There is also inconsistency in the way journals are referenced eg. compared references 23 and 24, both referring to Medical education. Please check that that actual reference source is made available in Reference 32.
As a reviewer, I enjoyed reading the paper which, on the whole, comprehensively describes the development of the processes and their implementation to practice.
Author Response
Thank you for your comments. We have edited the manuscript in accordance with your recommendations:
Reviewer 2:
In general, the paper was well presented and flowed logically with accurate description of the developments. As a reader, I did struggle to understand the process of ePortfolio review against the system based standards and would suggest that this needs described in further detail to aid understanding (194-197).
I feel that the paper would also benefit from a diagrammatic overview of the process, such as a flow chart. There is a lot of text used to describe the process and a diagram would help to summarise the processes and to also aid the understanding of those who prefer to learn visually.
Figure 3 Overview of ePortfolio Review has been added to summarise the process of ePortfolio Review. We hope that this summarises the process so that it is more accessible for the reader.
The paper does require review to ensure that it conforms with stylistic protocol. Examples that I have noted include:
(i) the abbreviation IIOP is used in line 80 without prior reference to the full name of the Centre.
This has been revised to ‘the Irish Institute of Pharmacy (IIOP)
(ii) in line 134, organization is spelt with a z whilst an s is used within Figure 1
The spelling has been revised within Figure 1 so that a z is now used consistently
(iii) within the references, the use of capital letters is sometimes missing eg. Reference 4, 15. There is also inconsistency in the way journals are referenced eg. compared references 23 and 24, both referring to Medical education. Please check that that actual reference source is made available in Reference 32.
References have been revised to ensure consistency